# Effects of *Loigolactobacillus coryniformis* K8 CECT 5711 on the Immune Response of Elderly Subjects to COVID-19 Vaccination: A Randomized Controlled Trial

**DOI:** 10.3390/nu14010228

**Published:** 2022-01-05

**Authors:** Anxo Fernández-Ferreiro, Francisco J. Formigo-Couceiro, Roi Veiga-Gutierrez, Jose A. Maldonado-Lobón, Ana M. Hermida-Cao, Carlos Rodriguez, Oscar Bañuelos, Mónica Olivares, Ruth Blanco-Rojo

**Affiliations:** 1Pharmacy Department, University Clinical Hospital of Santiago de Compostela (SERGAS), 15706 Santiago de Compostela, Spain; anxordes@gmail.com (A.F.-F.); roi.veiga.gutierrez@sergas.es (R.V.-G.); ana.maria.hermida.cao@sergas.es (A.M.H.-C.); 2Pharmacology Group, Health Research Institute Santiago Compostela (IDIS), 15706 Santiago de Compostela, Spain; 3Medicina Familiar y Comunitaria, 15706 Santiago de Compostela, Spain; fjformigo@hotmail.com; 4Research and Development Department, Biosearch Life, a Kerry Company, 18004 Granada, Spain; jamaldonado@biosearchlife.com (J.A.M.-L.); crodriguez@biosearchlife.com (C.R.); obanuelos@biosearchlife.com (O.B.); molivares@biosearchlife.com (M.O.)

**Keywords:** COVID-19, probiotic, immune response, elderly

## Abstract

Elderly people are particularly vulnerable to COVID-19, with a high risk of developing severe disease and a reduced immune response to the COVID-19 vaccine. A randomized, placebo-controlled, double-blind trial to assess the effect of the consumption of the probiotic *Loigolactobacillus coryniformis* K8 CECT 5711 on the immune response generated by the COVID-19 vaccine in an elderly population was performed. Two hundred nursing home residents >60 yrs that had not COVID-19 were randomized to receive *L. coryniformis* K8 or a placebo daily for 3 months. All volunteers received a complete vaccination schedule of a mRNA vaccine, starting the intervention ten days after the first dose. Specific IgG and IgA antibody levels were analyzed 56 days after the end of the immunization process. No differences between the groups were observed in the antibody levels. During the intervention, 19 subjects had COVID-19 (11 receiving K8 vs. 8 receiving placebo, *p =* 0.457). Subgroup analysis in these patients showed that levels of IgG were significantly higher in those receiving K8 compared to placebo (*p* = 0.038). Among subjects >85 yrs that did not get COVID-19, administration of K8 tended to increase the IgA levels (*p* = 0.082). The administration of K8 may enhance the specific immune response against COVID-19 and may improve the COVID-19 vaccine-specific responses in elderly populations.

## 1. Introduction

Since the coronavirus disease 2019 (COVID-19) was declared a global pandemic by the World Health Organization in March 2020, millions of cases and deaths have occurred [1], causing enormous human and economic costs worldwide. In this context, the development of vaccines was imperative, and, by the end of 2020, the first vaccines against severe acute respiratory syndrome coronavirus-2 (SARS-CoV-2) were approved by the National Competent Authorities, and the first vaccinations were performed [2,3].

The elderly population is particularly vulnerable to COVID-19, as a high risk of severe disease and hospitalization has been observed in elderly adults, and higher death rates have been reported in this population [4,5,6]. In addition, a lengthened time of hospitalization, delayed viral clearance, and a higher incidence of post-COVID-19 symptomatology were also related to advanced age [7,8]. Consequently, this population group was prioritized to receive the COVID-19 vaccine. However, although vaccinated individuals showed substantial protection against severe disease [9,10], several studies have shown a delayed and reduced immune response in elderly vaccinees compared to the youngest vaccinees [11,12,13]. This worrying fact, partly explained by the immunosenescence phenomenon [14], creates new questions about the necessity of finding specific solutions for enhancing the immune response of this vulnerable population due to SARS-CoV-2 mutations and the continuous spread of COVID-19 [12].

The former *Lactobacilli* genus, recently reclassified in 25 new genera, including *Loigolactobacillus* [15], has been demonstrated to enhance the immune response [16] and to increase the response to vaccines by acting as adjuvants [17,18]. In this regard, several studies have corroborated the immunomodulatory activity of the *Loigolactobacillus coryniformis* K8 CECT 5711 strain, both in adults [19] and in children [20,21]. Moreover, this strain has been shown to increase specific antibody levels against the hepatitis A virus when it was administered orally to healthy adults in the context of a hepatitis A vaccination [22]. More recently, a study performed in elderly subjects reported that the administration of *L. coryniformis* K8 increased the immune response against the influenza vaccine and decreased the symptoms associated with respiratory infections [23].

These data support the approach of using probiotic supplementation as adjuvants for boosting immunity and enhancing vaccine-specific responses in the elderly population [24,25] and corroborate the capability of the probiotic strain *L. coryniformis* K8 to enhance the immune response [22,23]. Herein, we report the findings of a randomized, placebo-controlled, double-blind trial to assess the effect of the consumption of the probiotic strain *L. coryniformis* K8 CECT 5711 on the immune response generated by the COVID-19 vaccine in an elderly population. The secondary aims were to evaluate the incidence of COVID-19 and the severity of the disease in the case of a COVID-19 outbreak.

## 2. Materials and Methods

### 2.1. Study Design and Subjects

A randomized, double-blinded, placebo-controlled multicenter study was performed. The study started in January 2021 and ended in April 2021.

Volunteers were recruited from three nursing homes with medical attention located in the province of A Coruña (Galicia, Spain). The inclusion criteria were nursing home residents older than 60 years who had received the first dose of a COVID-19 vaccination schedule and agreed to a blood extraction. The exclusion criteria included having a medical history of COVID-19 before the start of the intervention, presenting symptomatology compatible with COVID-19 at the beginning of the intervention, or being diagnosed with an immunocompromising condition. The study was conducted according to the Declaration of Helsinki, and the protocol was approved by the Regional Ethical Committee (Granada, Spain). Informed consent was obtained from all subjects. The trial was registered with the US Library of Medicine (http://www.clinicaltrials.gov, accessed on 1 December 2021) under the number NCT04756466.

The main outcome for the calculation of the sample size was the specific IgG antibody levels generated in response to the COVID-19 vaccine. Based on the data observed in older adults [26], a coefficient of variability of 0.76 was estimated for our sample. The sample size calculation was defined for the comparison of two independent samples with lognormal distribution [27]. For an alpha of 5% and a power of 80%, and taking 0.76 as the coefficient of variability and 35% as the minimum difference of interest to be detected between the groups, and considering a possible loss of 15% of subjects, a sample of 99 subjects per group (total *n* = 198) was necessary.

Finally, a total of 200 subjects were recruited. Volunteers were randomly assigned to one of two groups according to a randomization scheme generated by a computer program. All volunteers received two doses, 21 days apart, of the BNT162b2 mRNA COVID-19 vaccine (BioNTech/Pfizer). Ten days after the first dose, volunteers started the intervention (Appendix A in the Appendix A). Thus, the individuals in the placebo group consumed a capsule containing 220 mg of maltodextrin daily, whereas the individuals in the probiotic group consumed a capsule containing 3 × 10^9^ colony forming units of the *Loigolactobacillus coryniformis* K8 strain in a matrix of the same maltodextrin mixture daily. The probiotic and placebo were provided in identical gelatin capsules and packaged in identical plastic containers only differentiated by the randomization code. Treatments were given to the subjects for 3 months. All data about the health conditions of the volunteers and the consumption of medical treatments were evaluated and recorded monthly by a medical doctor in a case report form corresponding to each volunteer at baseline. Adverse events, defined as any unfavorable, unintended effects, were recorded at the follow-up visits (at 1, 2, and 3 months) (Appendix A in the Appendix A).

### 2.2. Study Outcomes and Sample Collection

The primary outcome of the study was to evaluate the immunological response to the vaccination by measuring the levels of antibodies against the SARS-CoV-2 spike protein. The secondary outcomes included analyzing cytokines related to the immune response, evaluating the incidence of SARS-CoV-2 infections confirmed by PCR or antigen testing, and determining the severity and duration of SARS-CoV-2 infections. The PCR or antigen test to determine SARS-CoV-2 infections was performed for the volunteers under medical criteria, for subjects with compatible symptomatology, or in the case of an outbreak in the nursing homes. All COVID-19 patients were medically followed, and daily symptoms related to the infection were recorded until the remission of the infection. The severity of SARS-CoV-2 infection was determined according to the National Institutes of Health (NIH)’s criteria [28] as asymptomatic, mild illness, moderate illness, severe illness, or critical illness. The duration of COVID-19 was determined as the time in days from the diagnosis until the complete resolution of the symptoms related to the infection. All COVID-19 patients continued taking the study product except those with critical illness.

Blood sample collection was performed 56 days after the end of the immunization process (second dose of the vaccination) (Appendix A in the Appendix A). Blood was collected in vacutainer tubes (BD Biosciences) and allowed to clot. Within an hour, the serum was separated by centrifugation at 1.000–1.500× *g* for 10 min, and serum aliquots were stored at −20 °C. The next day, the blood samples were cold-transported to Biosearch Life facilities in Granada (Spain), where the samples were processed and analyzed within 1 month of collection.

### 2.3. SARS-CoV-2-Binding IgG and IgA Antibody Levels and Cytokine Level Measurements

Quantitative measurements of human IgG and IgA antibodies against the RBD domain of the S1 protein (S1-RBD) of SARS-CoV-2 were determined using the RayBio COVID-19 S1 RBD protein Human (IgG or IgA) ELISA kit (RayBiotech, Peachtree Corners, GA, USA). Analyses were performed according to the instructions of the manufacturer.

Serum levels of interferon (IFN)-γ were measured by the ProQuantum™ Human IFN-γ Immunoassay Kit (Thermo Fisher Scientific, Rockford, IL, USA), whereas the serum levels of transforming growth factor (TGF)-β were quantitatively determined by an uncoated ELISA kit (Thermo Fisher Scientific, Rockford, IL, USA). The assays were performed according to the instructions of the manufacturer.

### 2.4. Statistical Analysis

The normality of the distribution for all measured variables was tested by normal probability plots and the Shapiro–Wilk test. Data are presented as the mean (standard deviation, SD) for continuous variables and as *n* (%) for categorical variables.

For comparisons between the groups at the beginning of the study (probiotic group vs. control group), continuous variables were analyzed with Student’s t test or the nonparametric Mann–Whitney U method, as appropriate, and categorical variables were analyzed with chi-square tests.

Data from the immunogenicity analysis are presented as the geometric mean and 95% confidence interval (95% CI) in the tables or as the mean (SE) of the log transformed data in the figures. Differences between the groups were evaluated by univariate model analysis, adjusted by the corresponding covariates. Testing for subgroup differences was performed when appropriate.

Data from the cytokine level analysis are presented as the mean (SD) in the tables or as the mean (SE) of the log transformed data in the figures. Differences between the groups were evaluated by univariate model analysis, adjusted by the corresponding covariates. Testing for subgroup differences was performed when appropriate. Linear regression statistical models to test for their association with IgG and IgA antibody levels, adjusted by age, sex, and disease index, were performed.

The occurrence of SARS-CoV-2 infections was described using the incidence ratio (IR) and incidence rate ratio (IRR) with the 95% CI and *p* value for the IRR by a logistic regression model. An ordinal logistic regression model was applied to evaluate the influence of the intervention on the severity of the infection. Differences in the duration of symptoms between the groups were determined by univariate model analysis. All the models were adjusted by age, sex, and disease index.

A general alpha level of 0.05 was used as the cutoff point for statistical significance. Statistical analysis was carried out using SPSS software version 27.0 for Windows (SPSS, Chicago, IL, USA).

## 3. Results

### 3.1. Study Data, Compliance and Baseline Characteristics of the Subjects

A total of 364 older adults were assessed for eligibility, of whom 164 were excluded for the following reasons: 96 refused to participate, 63 did not meet the inclusion criteria, and 5 passed away during the recruitment period. Finally, 200 volunteers were recruited and randomly distributed into two groups: the control group (*n* = 101) and the probiotic group (*n* = 99). Before completion of the 3-month intervention period, six volunteers in the control group and five in the probiotic group discontinued the intervention and dropped out of the study for the reasons detailed in the study flow chart (Figure 1).

No differences between the groups were detected between the number and causes of withdrawal. The compliance rate was confirmed to be very high (≈100%). Data were analyzed for all the subjects included in the study who had received the intervention for at least 10 days from the beginning of the intervention (analysis per intention to treat (ITT), *n* = 100 in the control group and *n* = 98 in the probiotic group). Immunogenicity analyses were performed for 193 volunteers (*n* = 97 in the control group and *n* = 96 in the probiotic group). No adverse events resulting from the intake of either type of treatment were reported.

The baseline characteristics of the 198 older adults included in the ITT group are presented in Table 1. No significant differences were detected between the subjects in the study groups except for suffering from cancer in the past (3% of the volunteers in the control group vs. 15.3% in the probiotic group, *p =* 0.003).

Appendix A in the Appendix A shows the medications prescribed to the volunteers due to their comorbidities before the intervention. No significant differences were detected among the subjects in the study groups.

### 3.2. COVID-19 Infection Incidence, Severity, and Duration

During the intervention and before receiving the second dose of the vaccine, 19 subjects were infected with the SARS-CoV-2 virus due to COVID-19 outbreaks in two of the nursing homes. The mean time between the beginning of the intervention and the date of COVID-19 diagnosis was 8.84 ± 4.20 days (10.37 ± 3.58 days in the control group vs. 7.72 ± 4.43 days in the probiotic group, *p =* 0.183). Of the 19 COVID-19 patients, 8 cases occurred in the control group (incidence rate (IR) (SD) = 0.08 (0.027)) and 11 cases occurred in the probiotic group (IR (SD) = 0.11 (0.032)). There was no significant difference between the groups in the incidence of COVID-19 infection (IRR (95% CI): 1.437 (0.552–3.741), *p =* 0.457). When the model was adjusted by sex, age, and the disease index, the IR (SD) in the control group was 0.06 (0.024) and 0.08 (0.030) in the probiotic group, with no significant difference between the groups (IRR (95% CI): 1.351 (0.530–3.441), *p =* 0.529).

Table 2 shows the details about the severity and duration of infection in the subjects who were diagnosed with COVID-19 infections. The percentage of asymptomatic patients in the probiotic group was higher than that in the control group (36.5% vs. 12.5%); however, the difference did not reach statistical significance (*p =* 0.189). The two patients classified as presenting “critical illness” (one per group) died 11 and 10 days after disease onset (no significant difference between the groups). Therefore, the time to symptom resolution was calculated in volunteers who survived (*n* = 7 in the control group and *n* = 10 in the probiotic group). There was no significant difference in the mean duration of COVID-19 between the control and probiotic groups (6.13 ± 4.22 and 8.45 ± 9.69 days, respectively, *p =* 0.587).

### 3.3. Immune Response to the COVID-19 Vaccine

Evaluations of IgG- and IgA-specific antibody levels against the S1-RBD protein of SARS-CoV-2 were performed independently using ELISA. Fifty-six days after the second dose of the COVID-19 vaccine, all volunteers presented levels of IgG above the detection limit, whereas four subjects in the control group and three subjects in the probiotic group presented levels of IgA below the detection limit (*p =* 0.711). No significant differences between the groups in the specific IgG levels (*p =* 0.552) or in the IgA levels (*p =* 0.422) were observed (Appendix A in the Appendix A).

Subgroup analyses, to determine the effect of the probiotic in the subjects who had a SARS-CoV-2 infection during the intervention, were performed (Appendix A in the Appendix A). Specific SARS-CoV-2 S1 RBD IgG levels were significantly higher in the subjects who received *L. coryniformis* K8 than in those who received the placebo (*p* = 0.038), whereas IgA levels were not different (*p* = 0.558) (Figure 2A).

Additionally, subgroup analyses by median age to determine the effect of the probiotic in the subjects who did not develop a SARS-CoV-2 infection during the intervention were performed (Appendix A in the Appendix A). The administration of *L. coryniformis* K8 tended to increase the specific SARS-CoV-2 S1 RBD IgA levels in subjects older than 85 years who did not develop a COVID-19 infection (*p =* 0.082), with no differences in IgG levels (*p =* 0.625) (Figure 2B).

Regarding cytokine levels, no significant differences were found between the probiotic and the control groups in any of the volunteers or in the subgroup analyses that were performed (Appendix A in the Appendix A). However, a significant relationship was found between the serum levels of SARS-CoV-2 S1 RBD IgA and the serum levels of TGF-β in the subjects who consumed the probiotic (β 2.291 per unit increase of TGF-β in logarithmic scale; *p =* 0.005) but not in the subjects who consumed the placebo (*p =* 0.929) (Figure 3). No relationships were observed between the levels of SARS-CoV-2 S1 RBD IgG and the levels of INF-g for the probiotic and control groups (*p =* 0.268 and *p =* 0.643, respectively).

## 4. Discussion

The elderly population is greatly affected by COVID-19, mainly due to the age-related changes affecting immune responses [29]. Vaccination has been proven essential for the reduction in the severity and mortality of older subjects [30,31], but it is known that age-related immune response defects provoke a lower level of protection after vaccination compared to younger subjects [11,12,13]. In this sense, several authors have highlighted the importance of using probiotic supplementation as adjuvants for boosting immunity and enhancing vaccine-specific responses in the elderly population [24,25]. However, to our knowledge, this is the first randomized clinical trial evidencing the usefulness of the administration of a probiotic strain in the context of the COVID-19 pandemic in elderly people by enhancing the specific immune response after COVID-19 infection and by helping vaccine-specific responses in the oldest population.

Increasing studies have demonstrated a higher immune response to mRNA COVID-19 vaccines in individuals previously infected with SARS-CoV-2 compared to antigen-naïve subjects [32,33,34]. The results obtained in this study support the observed evidence and add new information to the few data published regarding the elderly population [35,36]. Although this heightened immune response seems to be related to repeated exposure, the underlying mechanisms remain unclear [37]. Interestingly, in the subgroup of subjects that were diagnosed with COVID-19 during the intervention, the IgG-specific levels against the S1-RBD protein of SARS-CoV-2 were significantly higher in the patients who received the probiotic than in those who received the placebo two months after the beginning of the intervention. Cell wall components of probiotic lactobacilli, such as peptidoglycan and lipoteichoic acid, can trigger the innate immune response. Peptidoglycan, for example, is able to induce monocytes to express IL-1, IL-6, and TNF-alpha, leading to the stimulation of T or B cell proliferation and B cell differentiation [38]. Previous studies performed in murine bone marrow-derived macrophages showed the capability of *L. coryniformis* K8 in inducing the production of TNF-alpha (data not shown); however, no effects on these cytokines were observed in previous studies performed with this probiotic strain in humans [22,23]. More studies should be performed to elucidate the mechanisms involved in the activation of IgG production.

IgA is the predominant immunoglobulin in the respiratory tract and plays an important role in early defense and viral containment. The BNT162b2 mRNA COVID-19 vaccine induces both IgG and IgA production [39]. It has been reported that IgA has an active role against SARS-CoV-2 [40]; in fact, IgA has been described as contributing to virus neutralization to a greater extent than IgG and is probably associated with protection against reinfection [41]. In the present study, it was observed that the administration of the probiotic strain tended to increase the IgA response, which could help to increase protection against viral infection and, therefore, may have important clinical applications. Previous studies performed with fermented milk containing *L. coryniformis* K8 in combination with a *Lactobacillus gasseri* strain showed higher levels of IgA in feces and saliva [19,20,21]. In agreement with previous reports [42,43,44], and with the immunosenescence hypothesis [14], we found an inverse relationship between age and the immune response, with older volunteers who had not been infected with SARS-CoV-2 presenting a lower postvaccination immune response (data not shown). Thus, it is interesting that the effect of the probiotic on IgA production was mainly observed in the older population (>85 years old). The mechanism involved in the activation of the production of IgA by the probiotic strain could be related to the activation of TGF-β. Although no significant differences were observed in TGF-β levels between the control and probiotic groups, the fact that a significant positive correlation between IgA and TGF-β levels was only observed in the group receiving *L. coryniformis* K8 suggests a mediating role of TGF-β in the effect of the probiotic on IgA levels. The increase in IgA levels through the induction of TGF-β has been previously described for other strains [45,46]; however, further studies should be performed to clarify the mechanisms induced by *L. coryniformis* K8.

In addition, as vaccines simulate the contact of live viral antigens with the immune system [47], the better K8-induced response against the BNT162b2 mRNA COVID-19 vaccine might be translated to a better immune response against the live virus in the case of infection. Numerous reviews have proposed the possible role of probiotics in the mitigation of COVID-19 severity due to their immunomodulatory properties [48,49,50,51]. In the present study, 9.5% of the studied population was infected by the SARS-CoV-2 virus before receiving the second dose of the BNT162b2 mRNA COVID-19 vaccine. Although the percentage of asymptomatic patients was three times higher in the probiotic group than in the control group, we did not observe significant differences, probably due to the low number of observed cases and, thus, the loss of statistical power. In a previous study conducted with the *L. coryniformis* K8 in the context of the influenza vaccination, the probiotic strain was shown to decrease the incidence of symptoms associated with respiratory infections in an elderly population in a follow-up period of 6 months after the intervention. However, the strain was orally administered two weeks before contact with the viral antigens [23]. In the present study, the COVID-19 outbreaks occurred at the beginning of the intervention when the subjects had been taking the probiotic for less than 10 days. Therefore, we cannot discount that a significant beneficial outcome would have been observed if the infected subjects had taken the probiotic for a longer period of time before the COVID-19 outbreak. Further studies should be done to demonstrate this hypothesis.

Some limitations of this study must be acknowledged. First, the sample size in the subgroup analysis of the COVID-19-infected subjects was too small, therefore this result should be interpreted cautiously. Moreover, the study was performed in older Caucasian residents with several comorbidities, which limits the generalization of our results to healthy individuals and to other age groups or ethnicities. Therefore, further studies in other population groups should be performed.

## 5. Conclusions

In conclusion, the administration of *L. coryniformis* K8 CECT 5711 to the elderly population in the context of the COVID-19 vaccine enhanced the immune response in subjects who were previously infected with the SAR-CoV-2 virus and tended to improve the postvaccine immune response in the oldest subjects who were not infected with the virus. These results add evidence to previous clinical data [21,22] corroborating the capability of the probiotic strain *L. coryniformis* K8 to enhance the immune response. Probiotic administration may be a natural and safe strategy to improve the efficacy of vaccines, especially in vulnerable populations such as the elderly. Future studies should be performed to determine the role of probiotics in the prevention and mitigation of COVID-19 infection.

## Figures and Tables

**Figure 1 nutrients-14-00228-f001:**
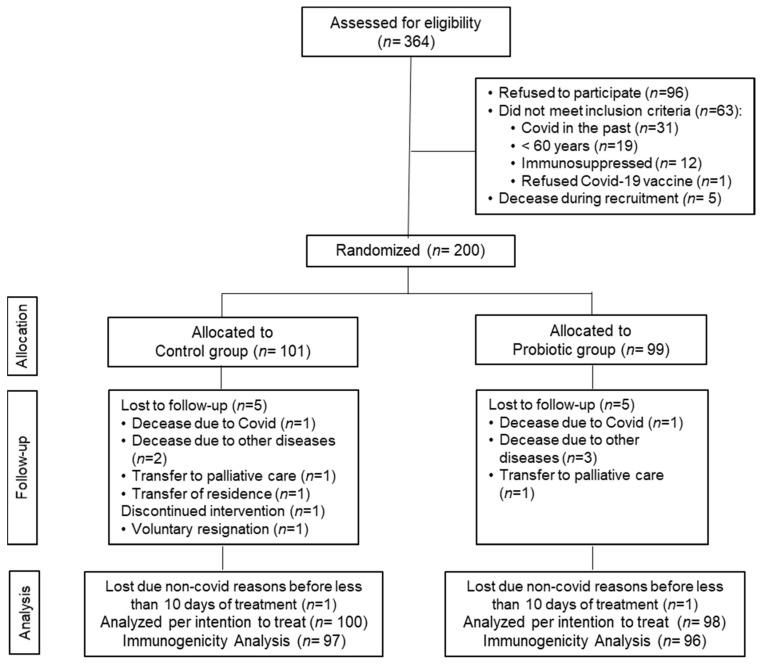
Flow chart of the study.

**Figure 2 nutrients-14-00228-f002:**
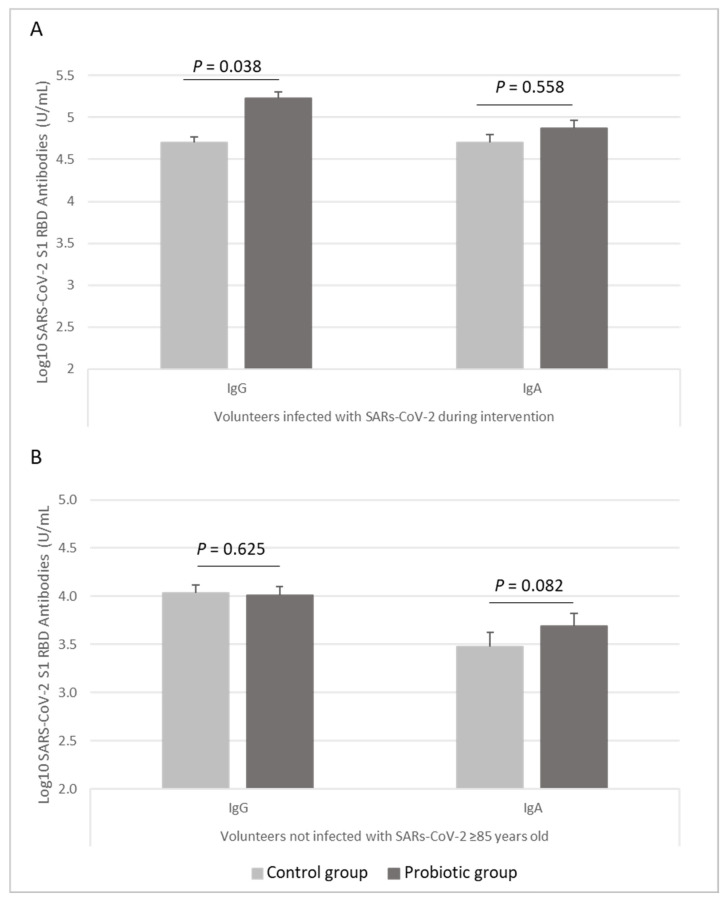
Levels of SARS-CoV-2 S1 RBD IgG and IgA antibodies (represented in Log10 of U/mL) in (**A**) volunteers infected with SARS-COV-2 during intervention (*n* = 10 in the probiotic group, *n* = 6 in the control group) and (**B**) volunteers not infected with SARS-CoV-2 older than 85 years old (*n* = 40 in the probiotic group, *n* = 48 in the control group). Data are represented as mean (bars) and SE (vertical lines). *p* value indicated differences between probiotic group (dark grey bars) and control (light grey bars) groups (univariate models adjusted by age, sex, disease index, and time to Covid-19 symptom’s resolution in Covid-19-infected subjects and adjusted by sex, disease index, and glucocorticoids in uninfected subjects).

**Figure 3 nutrients-14-00228-f003:**
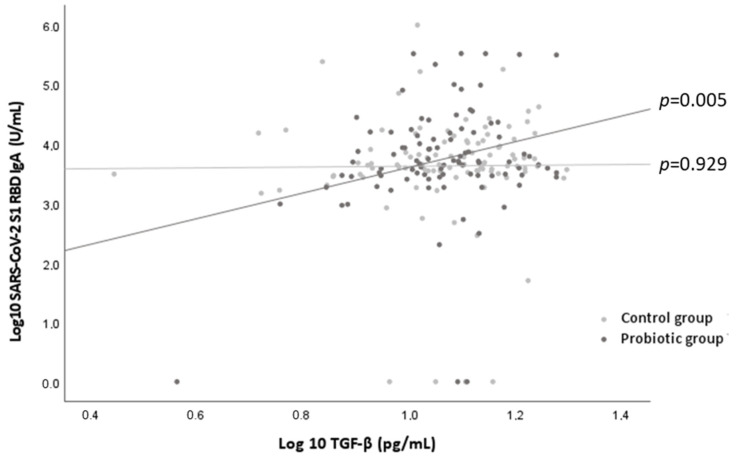
Relationship between the levels of SARS-CoV-2 S1 RBD IgA (represented in Log10 of U/mL) and levels of TGF-β (represented in Log10 of pg/mL) by control group (light grey points) and probiotic group (dark grey points). *p* values indicated the statistical significance of the linear regression analysis by intervention group adjusted by age, sex, and disease index.

**Table 1 nutrients-14-00228-t001:** Baseline characteristics of the subjects participating in the study.

		All Volunteers (*n* = 198)	Control Group (*n* = 100)	Probiotic Group (*n* = 98)	*p* between-Groups
Age (years)	Mean ± SD	83.13 ± 9.13	83.97 ±8.99	82.28 ± 9.22	0.192
	Median (IQR)	85 (77–90)	86 (77–90)	84 (77–90)	
Nursing home					0.992
Sta Olalla	*n* (%)	39 (19.7%)	20 (20.0%)	19 (19.4%)	
San Marcos	*n* (%)	91 (46%)	46 (46.0%)	45 (45.9%)	
San Simon	*n* (%)	68 (34.3%)	34 (34.0%)	34 (34.7%)	
Sex					0.356
Men	*n* (%)	73 (36.9%)	40 (40.0%)	33 (33.7%)	
Women	*n* (%)	125 (63.1%)	60 (60.0%)	65 (66.3%)	
Postural control					0.345
Stand upright	*n* (%)	138 (69.7%)	65 (65.0%)	73 (74.5%)	
Non-stand upright	*n* (%)	21 (10.6%)	12 (12.0%)	9 (9.2%)	
Bedridden patient	*n* (%)	39 (19.7%)	23 (23.0%)	16 (16.3%)	
BMI (kg/m^2^) ^1^	Mean ± SD	26.63 ± 5.11	26.25 ± 4.92	26.96 ± 5.28	0.411
Obesity	*n* (%)	42 (21.2%)	21 (21.0%)	21 (21.4%)	0.941
Low weight	*n* (%)	34 (17.2%)	15 (15.0%)	19 (19.4%)	0.413
Smokers	*n* (%)	18 (9.1%)	8 (8.0%)	10 (10.2%)	0.590
Former alcoholics	*n* (%)	10 (5.1%)	4 (4.0%)	6 (6.1%)	0.495
Dyslipidemia	*n* (%)	80 (40.4%)	38 (38.0%)	42 (42.9%)	0.486
Hypertension	*n* (%)	125 (63.1%)	63 (63.0%)	62 (63.3%)	0.969
Diabetes Mellitus	*n* (%)	47 (23.7%)	23 (23.0%)	24 (24.5%)	0.805
Cardiovascular Disease	*n* (%)	77 (38.9%)	41 (41.0%)	36 (36.7%)	0.538
Cognitive Diseases ^2^	*n* (%)	91 (46.0%)	47 (47.0%)	44 (44.9%)	0.767
Chronic Lung Disease	*n* (%)	27 (13.6%)	15 (15.0%)	12 (12.2%)	0.572
Renal disease	*n* (%)	21 (10.6%)	12 (12.0%)	9 (9.2%)	0.520
Hepatic disease	*n* (%)	7 (3.5%)	4 (4.0%)	3 (3.1%)	0.721
Previous cancer diagnosis	*n* (%)	18 (9.1%)	3 (3.0%)	15 (15.3%)	0.003
Rheumatic diseases ^3^	*n* (%)	57 (28.8%)	27 (27.0%)	30 (30.4%)	0.562
Psychiatric diseases ^4^	*n* (%)	82 (41.4%)	36 (36.0%)	46 (46.9%)	0.118
Disease Index ^5^	Mean ± SD	3.20 ± 1.62	3.09 ± 1.66	3.32 ± 1.58	0.328
	Median (IQR)	3 (2–4)	3 (2–4)	3 (2–4)	
Number of habitual medications	Median (IQR)	8 (5–11)	7.5 (5–10)	9 (6–11)	0.108

Values are mean ± SD and/or median (IQR) for continuous variables and *n* (%) for categorical variables. *p* indicates differences between the control group and the probiotic group ^1^. BMI were calculated in volunteers that stand upright ^2^. Cognitive Diseases included Alzheimer, Parkinson, and Dementia ^3^. Rheumatic diseases not under immunosuppressive therapies ^4^. Psychiatric diseases included bipolar disease, major depression, and schizophrenia ^5^. Disease Index included the sum of HTA, T2DM, CVD, Lung diseases, oncology disease in the past, rheumatology diseases, cognitive diseases, psychiatric diseases, renal diseases, hepatic diseases, and dyslipidemia.

**Table 2 nutrients-14-00228-t002:** Classification of severity and duration of the infection in Covid-19 patients by intervention group.

		All Patients (*n* = 19)	Control Group (*n* = 8)	Probiotic Group (*n* = 11)	*p* between-Groups
Classification of severity					0.189
Asymptomatic Infection	*n* (%)	5 (26.3%)	1 (12.5%)	4 (36.4%)	
Mild Illness	*n* (%)	3 (15.8%)	2 (25%)	1 (9.1%)	
Moderate Illness	*n* (%)	2 (10.5%)	1 (12.5%)	1 (9.1%)	
Severe Illness	*n* (%)	7 (36.8%)	3 (37.5%)	4 (36.4%)	
Critical Illness	*n* (%)	2 (10.5%)	1 (12.5%)	1 (9.1%)	
Time to symptom resolution (days) ^1^	Mean ± SD	7.47 ± 1.79	6.13 ± 4.22	8.45 ± 9.69	0.587

Values are mean ± SD for continuous variables and *n* (%) for categorical variables. *p* indicates differences between the control group and the probiotic group (ordinal logistic regression test for the categorical variable and univariate test for continuous variable) adjusted by age, sex, and the disease index^1^. Time to symptom resolution was calculated in volunteers that survived (*n* = 7 in the control group and *n* = 10 in the probiotic group).

## Data Availability

Data sharing is not applicable to this article.

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
