# Peer review of "Effects of Loigolactobacillus coryniformis K8 CECT 5711 on the Immune Response of Elderly Subjects to COVID-19 Vaccination: A Randomized Controlled Trial"

_nutrients, 2022, doi:10.3390/nu14010228_

Round 1

Reviewer 1 Report

Fernández-Ferreriro, et al., assessed the effect of the consumption of the probiotic Loigolactobacillus coryniformis K8 CECT 5711 on the immune response generated by the COVID-19 vaccine in an elderly population. They  analyzed specific IgG and IgA antibodies and cytokines (INF-g and TGF-β) 56 days after the end of the immunization process and concluded that the administration of K8 may enhance the specific immune response against COVID-19 and may improve the COVID-19 vaccine-specific responses in elderly populations.

The manuscript is interesting, but some concerns need to be addressed. For example, as mentioned in lines 80-81, being diagnosed with an immunocompromising condition should be excluded from the study. However, in the table 1 there are subjects with different diseases such as Rheumatic diseases (about 29%) and it is not clear if they are under the immunosuppressive therapies. Also, if possible, the limitations of the study should be addressed.

Reviewer 2 Report

The article is well written, but I have some comments to make. 

-On line 148-149 you say you used the Kruscall test, but when in the presence of two groups the Kruskal test is simply the Mann U Whitney test. So instead of Kruskal test, use Mann U Whitney test.

- On line 229 you say you only have 7 patients. Obviously this is not a number that defines a representative sample, thus any test performed with a group of 7 patients is the result of a not very serious analysis.

In Figure 2 you show a significant p-value computed between groups consisting of 6 and 10 patients. This result is absolutely inconclusive. Therefore, surely a bar plot is useful for the purpose of the experiment but a significant p-value distorts the literature because it is the result of a biased analysis. 

Thus, I suggest that within the limitations of the study, such reflections are included.
